# Blood Platelet Adenosine Receptors as Potential Targets for Anti-Platelet Therapy

**DOI:** 10.3390/ijms20215475

**Published:** 2019-11-03

**Authors:** Nina Wolska, Marcin Rozalski

**Affiliations:** Department of Haemostasis and Haemostatic Disorders, Chair of Biomedical Science, Medical University of Lodz, 92-215 Lodz, Poland; n.m.wolska@gmail.com

**Keywords:** adenosine, adenosine receptor, adenosine receptor agonist, platelet, AR, therapy

## Abstract

Adenosine receptors are a subfamily of highly-conserved G-protein coupled receptors. They are found in the membranes of various human cells and play many physiological functions. Blood platelets express two (A_2A_ and A_2B_) of the four known adenosine receptor subtypes (A_1_, A_2A_, A_2B_, and A_3_). Agonization of these receptors results in an enhanced intracellular cAMP and the inhibition of platelet activation and aggregation. Therefore, adenosine receptors A_2A_ and A_2B_ could be targets for anti-platelet therapy, especially under circumstances when classic therapy based on antagonizing the purinergic receptor P2Y_12_ is insufficient or problematic. Apart from adenosine, there is a group of synthetic, selective, longer-lasting agonists of A_2A_ and A_2B_ receptors reported in the literature. This group includes agonists with good selectivity for A_2A_ or A_2B_ receptors, as well as non-selective compounds that activate more than one type of adenosine receptor. Chemically, most A_2A_ and A_2B_ adenosine receptor agonists are adenosine analogues, with either adenine or ribose substituted by single or multiple foreign substituents. However, a group of non-adenosine derivative agonists has also been described. This review aims to systematically describe known agonists of A_2A_ and A_2B_ receptors and review the available literature data on their effects on platelet function.

## 1. Introduction

Activation of blood platelets plays a critical role in the pathogenesis of arterial thrombotic diseases, such as coronary heart disease, myocardial infarction, and stroke, which are the primary cause of mortality in developed countries. Therefore, anti-platelet therapy is one of the most important tools in the treatment of arterial thrombotic disorders [1].

Platelets express two receptors for ADP: The P2Y_1_ receptor, which initiates platelet aggregation, and the P2Y_12_ receptor, which enhances this process, finally leading to thrombus formation. In contrast to the P2Y_1_ receptor, the P2Y_12_ receptor is almost exclusively expressed in the platelet plasma membrane. Therefore, P2Y_12_ has become a major therapeutic target to prevent arterial thrombotic disorders instead of adenosine receptors [2]. In general, the major clinically approved P2Y_12_ inhibitors include the thienopyridine-class inhibitors (ticlopidine, clopidogrel, and prasugrel), the ATP analogue—cangrelor, and the cyclo-pentyl-triazolo-pyrimidine (CPTP)—ticagrelor [2,3]. Thienopyridines are prodrugs that are converted to short-living active metabolites; these irreversibly inactivate the receptor and consequently inhibit ADP-induced platelet activation. Cangrelor is the first intravenous P2Y_12_ receptor inhibitor to reversibly block ADP signaling in a non-competitive manner. Ticagrelor is an allosteric antagonist of P2Y_12_, acting directly via reversible binding to the P2Y_12_ receptor, which leads to the non-competitive inhibition of ADP-induced P2Y_12_ activation and is used for the prevention of thromboembolic events in patients with acute coronary syndromes [2,3,4,5]. As regards current clinical practice, clopidogrel, prasugrel, and ticagrelor are the most frequently used oral platelet P2Y_12_ inhibitors; the use of ticlopidine has been abandoned. Clopidogrel is the only oral P2Y_12_ inhibitor recommended for the treatment of patients with stable coronary artery disease. Although all three agents have an indication for use in acute coronary syndromes, current guidelines suggest the preferential use of prasugrel and ticagrelor over clopidogrel because of their superior clinical benefits, i.e., the improved efficacy, lowered individual variation in response, and less frequent and severe side effects [6]. Cangrelor, in turn, as the recently approved, first P2Y_12_ inhibitor administered intravenously, seems to be the most promising in percutaneous coronary interventions [7].

Although these anti-platelet agents are now commonly used as clinically approved drugs, effective therapy of arterial thrombosis still presents a problem. For example, gastrointestinal bleeding is a common adverse event observed in 5 to over 10% of patients treated with oral anti-platelet drugs. Many of the patients with this complication require recurrent hospitalization [8]. Another severe and relatively common side effect of anti-platelet therapy is a higher risk of intracranial and intracerebral hemorrhage [9]. On the other hand, for some patients, the applied anti-platelet therapy appears insufficient and does not prevent excessive clotting. This can be explained by the fact that anti-platelet agents either interfere with only one out of several pathways of platelet activation or, even if they block effectively a final common step of platelet aggregation, such as fibrinogen binding (blockers of fibrinogen receptor), their use is associated with a risk of bleeding [3]. Another problem affecting the efficiency of many anti-platelet drugs stems from the individual variability of the response to these drugs resulting from both environmental and genetic factors, especially in case of prodrugs [10]. Altogether, there still is a need for the development of novel platelet inhibitors with better efficacy and safety, or using a combined therapy based on various sets of currently-available agents.

Adenosine is an important purine metabolite, serving not only as a component of nucleic acids and the most important energy carrier in the cell—ATP—but also as a signaling molecule regulating tissue function [11,12]. Adenosine receptors (AR) are present in membranes of many types of human cells and play various physiological functions. Blood platelets express two (A_2A_ and A_2B_) of the four known adenosine receptor subtypes (A_1_, A_2A_, A_2B_, and A_3_). As regards platelet AR receptors, A_2A_ is characterized by the higher affinity to adenosine in comparison with A_2A_; furthermore, platelets have a significantly lower density of A_2B_ [13,14]. Activation of platelet AR results in an enhanced intracellular cAMP level and consequently leads to the inhibition of platelet activation and aggregation [15,16]. Therefore, adenosine receptors A_2A_ and A_2B_ could be considered as targets for anti-platelet therapy, especially under circumstances when classic therapy based on antagonizing the P2Y_12_ purinergic receptor is insufficient or problematic.

The aim of this review is to systematically present current knowledge of the impact of synthetic, selective, longer-lasting agonists of A_2A_ and A_2B_ receptors on platelet function inhibition, and evaluate their potential as anti-platelet therapeutics.

## 2. The Classification, Distribution, and Signaling of Adenosine Receptors

Adenosine receptors (AR) represent a subfamily of highly-conserved G-protein coupled receptors. They are found in membranes of various human cells and play a plethora of physiological functions. Four AR subtypes are known: A_1_, A_2A_, A_2B_, and A_3_. The A_1_ and A_3_ receptors preferentially couple to G_i_ protein to inhibit adenylate cyclase and, consequently, the production of cyclic AMP (cAMP). The A_2A_ and A_2B_ subtypes stimulate the production of cAMP by coupling to G_s_ or G_o_ protein [17]; they are therefore classified as adenylyl cyclase inhibiting (A_1_ and A_3_) or adenylyl cyclase activating (A_2A_ and A_2B_) [18].

AR subtypes are characterized by high resemblance in terms of amino acid sequence: The human A_1_ and A_3_ ARs are identical in 49%, whereas human A_2A_ and A_2B_ AR sequences are identical in 59%. In general, an AR molecule consists of a single polypeptide chain that transverses the membrane from the extracellular side, beginning at the N terminus and forming seven transmembrane helices [15]. AR receptors are commonly expressed in many tissues and cells types; however, the distribution of subtypes is highly tissue-specific (Table 1).

Adenosine receptors play multiple functions. A_2A_ receptor agonization is known to cause coronary artery vasodilatation, decreased dopaminergic activity in central nervous system, and inhibition of central neuron excitation, whereas A_2B_ receptor activation may cause bronchospasm [12,24,25,26]. Therefore, all adenosine receptor ligands should be used only with the utmost caution [11], despite some having already been approved for human use (one of them—regadenoson—is discussed further in this article). An interesting insight into adenosine receptor overstimulation may be gained from the study of adenosine deaminase deficiency—a rare, autosomal metabolic disorder that causes severe combined immunodeficiency [27]. In this syndrome, platelet dysfunction has been described, as well as severe thrombocytopenia [28,29]. It is, however, important to remember that there is no exact parallel between patients with this syndrome and an anti-platelet therapy with the use of AR agonists. The dose of synthetic adenosine agonist equivalent to adenosine would be much lower and, most importantly, it would be applied in adults.

As it has been already mentioned, blood platelets express two subtypes of AR receptors (A_2A_ and A_2B_); however, the expression (the number of receptor copies in the plasma membrane) of the two receptor types has not been established. A_2A_ receptor is believed to be expressed on platelets in higher density as compared to A_2A_ [18]. Only one study has estimated the gene expression profile for A_2A_ and A_2B_ in human platelets. This report demonstrated comparable mRNA expression levels for A_2A_ and A_2B_ AR [22], but no further evidence exists regarding protein levels present on the platelet surface. Moreover, two studies [30,31] have not been able to quantify A_2A_ AR in platelet proteome, while have easily identified, for example, P2Y_12_ receptor, expressed on healthy platelets in around 450–1000 copies [32].

A_2A_ AR was identified as an important receptor on platelets and a mediator of adenosine inhibition of platelet aggregation [33]. This is achieved through inhibition of mobilization of internal calcium stores and influx of external calcium, both associated with activation of adenylate cyclase and increase of cAMP concentration [34]. Cyclic nucleotides are also strong inhibitors of the release of calcium ions into the cytosol, which underpins many events in platelet activation. In addition to inhibiting platelet aggregation in human blood, the activation of A_2A_ AR by specific agonists leads to a reduction in P-selectin expression on the platelet cell surface, as a result of thromboxane A2 or ADP stimulation [15].

Phenotypically, counts of blood cell populations, including platelets, were found to be similar in A_2A_ AR knock-out mice and with wild-type mice [20]. In this knock-out model, the rate of ADP-induced platelet aggregation differed in both the genetic variants following the treatment with nonselective AR agonist, 5′-N-ethyl-carboxamidoadenosine (NECA). NECA administration led to inhibition of platelet aggregation in wild-type mice, but demonstrated no effect in A_2A_ AR-null mice [35].

The role of A_2B_ AR in platelets remains disputable. It was proposed that this AR subtype activates signal transduction pathways other than adenylate cyclase [36]. It was also proposed, based on a mouse knock-out study, that A_2B_ AR is upregulated under stress in vivo, and plays a significant role in regulating ADP receptor expression [23]. The same study also found that agonization of this receptor inhibits agonist-induced platelet aggregation, but it should be noted that no specific agonist was used: A combination of a non-selective agonist and A_2A_ receptor inhibitor was applied.

The half-life of adenosine in circulation is extremely short (approximately 1 s), due to the action of enzymes like adenosine deaminase, which convert it to inosine, or adenosine kinase, which phosphorylates it to 5′-AMP, or due to uptake by nucleoside transporters [33]. Therefore, close study and pharmacological potential of ARs can be facilitated only by finding longer-lasting synthetic agonists and antagonists.

## 3. Adenosine Receptor Agonists—Structure, Chemical Properties, and Known Effects on Platelet Function

The purpose of synthesizing novel AR agonists is to achieve longer-lasting agonization and selectivity between receptor subtypes without compromising high affinity of binding to the receptor. This is accomplished, with varying success, either by introducing additional substituents to the molecule of adenosine in the hope of improving a receptor-ligand binding, or by utilizing molecules of other chemical structure. A nomenclature and the chemical structure of AR agonists is presented in Table 2.

### 3.1. Adenosine Derivatives

Numerous adenosine-derived compounds have been tested for their affinity and selectivity for ARs. The most prevalent strategies for obtaining novel derivatives were substitutions of adenosine at the 2-position, usually with (thio)ethers, secondary amines, and alkynes, as well as at the N^6^-position. The latter substitutions appear to increase affinity to the A_2A_ receptor subtype [37]. Below, we present an overview of adenosine derivatives classified according to the position of substituents. It is, however, important to point out that the citied study was performed using various protocols and utilizing nonhomogeneous materials, hence is not possible to make a direct comparison between available data, for example for IC_50_ or EC_50_ measured for various AR agonists.

#### 3.1.1. Adenosine Derivatives with Substituents at C1 to C8 Positions

##### 2-chloroadanosine

2-chloroadenosine is one of the first characterized AR agonists, first described in 1964 [38,39]. Conducted studies were predominantly concentrated on its effect on platelets, from the pioneering research into activation signaling [40,41], including recognition of AR subtypes [42], through examination of platelet disorders [43,44], to investigations of 2-chloroadenosine antiaggregatory effects [45].

IC_50_ of 2-chloroadenosine for human platelets was established at 1.6 µM (CI95% 0.61–4.4) (by photometric method in PRP (platelet rich plasma) with ADP), while its EC_50_ was found to be 1.7 µM in adenylate cyclase assay using human platelet membranes (CI95% 1.5–2.0) [46]. For aggregation in whole blood, IC_50_ was later measured to be 2.3 µM [47]. 2-chloroadenosine is a non-selective AR agonist, with high affinity, especially to A_1_ and A_2_ AR classes [48,49]. Nowadays, it is employed in numerous research areas as a stable adenosine analogue [50,51,52,53]. Its advantage over the adenosine arises from the fact that 2-chloroadenosine has a longer half-time and exerts more potent activating effect on AR (A_2A_ receptor biding affinity Ki = 180 nM), being only minimally different chemically [54].

##### Regadenoson

Regadenoson is also commonly denoted as CVT-3146 and known under its trade names Lexiscan or Rapiscan. It is a selective A_2A_ AR agonist of low affinity (Ki = 1095 nM). It also binds to A_1_ receptor subtype (Ki > 16,460 nM), but has much higher affinity constants in the case of A_2B_ and A_3_ receptor classes [55]. It was approved by the Food and Drug Administration (FDA) in 2008 for diagnostic purposes in radionuclide myocardial perfusion imaging, manufactured by Astellas Pharma and marketed by GE Healthcare. It is administered intravenously in bolus as a 0.08 mg/mL solution. Regadenoson rapidly increases coronary blood flow to over twice the baseline value in 30 s and decreases to below twice the baseline value in 10 min, and is removed from the human body (58% through renal excretion) within two hours [56] (clinical studies: NCT01019486 (RABIT1D) and NCT00881218). The influence of regadenoson on platelet aggregation has not been reported in literature so far. However, our unpublished results (manuscript currently under review) have confirmed that regadenoson has an anti-platelet effect—in whole blood aggregation, the obtained maximal inhibition value was of 38.1 ± 3.2%, and IC_50_ of 1.2 µM.

##### Binodenoson

Binodenoson is another AR agonist currently approved for human use. It was firstly reported in the literature in 1996 under the name WRC-0470 as a short-acting A_2A_ agonist [57], then a year later, it was presented as a potential imaging tool [58]. Subsequently, it was tested specifically for induction of pharmacological stress as an adjunct to myocardial perfusion imaging. Its pharmacokinetics and safety profile were tested in clinical trials [59,60,61]. Binodenoson has successfully completed two phase III clinical trials (identifiers NCT00944294 and NCT00944970) and is currently used as a single bolus injection prior to myocardial perfusion imaging.

It is characterized with good selectivity for A_2A_ AR receptor over other AR receptors, and good binding affinity (Ki = 270 nM) [62]. Despite being well characterized concerning general safety, no data concerning platelets or its potential anti-platelet effect are available.

##### PSB Family

PSB-0777 was developed in PharmaCenter Bonn, and described in 2011 as a potential anti-inflammatory agent for a treatment of inflammatory bowel disease [63]. Despite it being a polar and water-soluble substance, it is not absorbable when administered *per os*, but suitable for parenteral application only. It was determined to be a full A_2A_ agonist, of high affinity (Ki = 44.4 nM) and high selectivity (over 225-fold) over other ARs. The compound exhibits affinity for both human and rat A_2A_. In cAMP accumulation assay using CHO cells expressing the A_2A_ receptor, EC_50_ was established at 117 nM. PSB-0777 was successfully utilized as an A_2A_ receptor agonist in a study concerning activation of brown adipose tissue [64]. It has yet to be investigated in the context of blood platelets.

A recent study reported the anti-platelet effects of three other recently-synthesized compounds from this family: PSB-15826, PSB-12404, and PSB-16301 [65,66,67,68]. PSB-15826 was found to be the most potent agonist out of these three compounds, characterized by IC_50_ values of 0.32 ± 0.05 µM for inhibition of platelet aggregation, 0.062 ± 0.2 µM for inhibition of platelet activation, and 0.24 ± 0.01 µM for cAMP production, making it a stronger anti-platelet agent than adenosine. PSB-16301 has also effectively reduced ADP-induced platelet aggregation with relatively low IC_50_ of 5.5 ± 0.2 µM, as well as PSB-12404, though at higher concentration: IC_50_ of 66.8 ± 0.07 µM [68]. Other members of this family were also described in the literature. They are either very weak AR agonists, or even AR inhibitors.

There are no data available on the cytotoxicity of this group of compounds; however, success in identifying multiple members of this family with anti-platelet properties suggests a high chance for finding an analogue with good safety profile.

##### MRE0094

MRE0094 is also known and marketed under names Sonedenoson and 2-[2-(4-chlorophenyl)ethoxy]adenosine. It is 39,000-fold more selective for adenosine A_2_ receptors than adenosine A_1_ receptors, with Ki for A_2_ AR subtype established at 490 nM [69]. It was successfully utilized as an A_2A_ selective compound in cell signaling research focused on varying topics [70,71,72], but most promising investigations concentrated on its use in promoting wound healing [73,74]. In early 2000, it was being developed by King Pharmaceuticals with hopes of becoming a novel topical drug.

MRE0094 was tested in two Phase II clinical trials concerning wound healing in chronic, neuropathic, diabetic foot ulcers, both of them sponsored by Pfizer. The first trial (ClinicalTrials.gov Identifier: NCT00312364) was completed in 2006; however, no results are available. The second study (ClinicalTrials.gov Identifier: NCT00318214) has been terminated due to poor enrolment of participants. According to King Pharmaceuticals, MRE0094 did not show expected improvement over selected reference for the clinical endpoints. MRE0094 has not been examined for its anti-platelet activity to date.

##### CV1808

CV1808 is also sometimes denoted as 2-phenylaminoadenosine. It is one of the first characterized AR agonists that was utilized in studies aiming to define AR subclasses [75,76,77]. CV1808 is a non-selective agonist, with Ki values of 560–1100 nM for A_1_, and 190 nM for A_2A_ [78]. It was later reported that A_2B_ Ki is similarly low to that of A_2A_ AR subtype [79].

CV1808 is being used in various investigations, mainly concerning cardiovascular and immune research areas [80,81,82,83,84]. CV1808 has yet to be tested for its anti-platelet properties.

##### AMP597

AMP597 was first described by Smits et al. in 1998 as a novel cardioprotective A_1_/A_2_ agonist [85]. It has high affinity for the A_1_ (Ki = 2 nM) and A_2A_ (Ki = 56 nM) receptor subtypes [86] and was later determined to be an A_2B_ agonist as well, based on the observation of its ability to induce phosphorylation of extracellular signal-regulated kinase and its protection against infarction in rabbit heart reperfusion studies [87]. It could be regarded, therefore, as a potent but non-selective AR agonist.

It has not been extensively studied; although it has been the subject of cardiac protection studies, a lack of publications since 2010 suggests that this line of research has been abandoned, despite the fact that it was undergoing clinical phase II studies in patients suffering acute myocardial infarction in 2000 [86]. Its effect on platelets remains unknown.

#### 3.1.2. Adenosine Derivatives with Substituents at C1′ to C5′ Positions

##### NECA

5’-N-ethylcarboxamidoadenosine, commonly abbreviated as NECA, was first described in 1977 as a vasodilator, and then as a platelet function inhibitor in the 1980s [88]. NECA was employed in early radioligand studies to characterize AR platelet receptors, and was established to bind to two distinct binding sides at submicromolar concentrations [89,90].

NECA IC_50_ for platelet aggregation in human material was established at 0.36 µM (CI95% 0.35–0.38 µM) [91]. Ki for human AR subtypes were set at 560 nM (480–650 nM) for A_1_, 620 nM (300–1300 nM) for A_2A_, and 6.2 nM (5.1–7.5 nM) for A_3_, showing a lack of selectivity between A_2A_ and A_1_ receptor subtypes [92]. However, it was described as a suitable A_2B_ agonist, with IC_50_ of 3.1 μM (cAMP production in CHO cells) [93].

NECA has no current medical applications and has never been a subject of clinical testing. It is most commonly used in basic, platelet and vascular, and neurological research.

#### 3.1.3. Compounds with Substituents at C1 to C8 and C1′ to C5′ Positions

##### CGS 21680

CGS 21,680 is one of the earliest synthesized adenosine analogue AR agonists. It was primarily used to elucidate the AR subclass division into A_2A_ and A_2B_ [76,94]. It is a strong, full agonist, selective towards A_2A_ (Ki A_2A_ = 27 nM, Ki A_2B_ is over1000 nM) [95]. It is probably the most commonly-employed AR A_2A_ agonist; it is used through a variety of research, especially in neurological studies. However, it has not been a subject of any clinical trials.

Its effect on platelets has been already established. Early studies reported IC_50_ 0.82 µM (CI95% 0.6–1.1) for human platelet aggregation, as measured by turbidimetry, and EC_50_ 0.083 ± 0.005 µM for stimulation of adenylate cyclase in human platelets [91]. Subsequently, CGS 21,680 has been used for further platelet research, including, but not limited to, studies on the association between depression and platelet signaling dysregulation [96], species-dependent platelet function [97], and neutrophil involvement and signaling in thrombosis [98].

##### HE-NECA

Another AR agonist extensively employed in a variety of research areas is the A_2A_ selective agonist HE-NECA, which was derived from non-selective AR agonist NECA. It has good selectivity between A_2_ and A_1_ receptor subclasses, but only slight selectivity between A_2_ and A_3_ subclasses (Ki A_2_ = 130 nM; Ki A_2_ = 2.2 nM; Ki A_3_ = 24 nM) [99]. It was reported as an anti-platelet agent in 1994 [99], when its anti-aggregatory potency was estimated as three-fold stronger in comparison to NECA. The anti-platelet activity of HE-NECA was confirmed in an in vivo study in rabbits, in which the drug was administered at a dose of 10 µg/kg: Platelet accumulation in pulmonary microcirculation was found to fall by over 50% after challenge with ADP [100]. HE-NECA was also used in a Borea group study investigating the influence of caffeine (AR inhibitor) on platelet function. In this study, HE-NECA was found to increase cAMP levels with an EC_50_ of 59 ± 3 nM, and inhibit ADP-induced human platelet aggregation (measured by turbidimetric method) with an IC_50_ of 90 ± 6 nM [101,102]. Recently, HE-NECA was also used in work focused on a quantification of different imaging approaches to experiments carried out under flow conditions, where 10 µM HE-NECA was found to inhibit clot formation under flow in whole blood by 82%, based on the volumes of aggregates recorded by confocal microscopy [103].

HE-NECA is also employed in other research disciplines, including renal function investigation [104], and neurological [105,106] and immunological [107] research.

##### UK-432094

This AR agonist is usually known as UK-432,094 in the literature, but notations UK-432094 or UK432094 are also in use. UK-432,094 was tested by PFIZER in a phase II clinical trial (ClinicalTrials.gov Identifier: NCT00430300) as an inhalation agent for severe chronic obstructive pulmonary disease; however, the trial was terminated due to low treatment effectiveness. UK-432094 is a selective A_2A_ agonist (Ki of 4.75 nM) [108], with reported EC_50_ as low as 5.4 ±1.8 nM (cAMP level evaluation in CHO cells stably expressing human A_2A_ and A_2B_ receptors) [109]. Its anti-platelet effect has been recently assessed using multiple electric aggregometry in whole blood. Using this technique, IC_50_ was found to be below 1 µM, with an inhibition rate of 40% at this concentration. The agonist had an ability to practically abolish aggregation at higher concentrations (79% inhibition at 100 μM), while demonstrating no cytotoxic effect on platelets [110].

UK-432,094 has been prevalently used in basic research of the receptor–ligand binding mechanisms, providing insights into A_2A_ receptor structure [108], binding sight dynamics and agonist efficacy [111,112,113], and receptor interactions with other molecules [114].

### 3.2. Non-Adenosine Compounds

Apart from compounds based on the adenosine molecule modified chemically by introducing various substituents, AR agonists could also be found among substances of different chemical structure. The examples of such the AR agonists are given below.

#### 3.2.1 BAY 60-6583

BAY 60-6583 was patented in 2001 as a highly-selective A_2B_ agonist (Ki A_2A_ is over10000 nM, Ki A_2B_ = 3–10 nM). Since then, it has been used in various research areas, including in vitro and in vivo immunological [115], cardiological [116,117], and oncological [118,119] research, lung disease and damage control studies [120,121], as well as the therapy of renal nephropathy [122]. So far, it is the only selective A_2B_ agonist in wide use. It has been reported to be a subject of pre-clinical studies to treat coronary artery disease and atherosclerosis [123], but it has not been registered for clinical trials.

The anti-platelet effects of BAY 60-6583 have not been studied in humans; however, Bot et al. report no decrease in aggregation (as measured by turbidimetry) or any change in platelet surface activation markers after treatment with BAY 60-6583 (50 µg/day) in ApoE^−/−^ mice [124].

#### 3.2.2 LUF5834 and LUF5835

A series of non-adenosine compounds were synthesized in 2004 as an attempt to generate an A_2B_ AR subtype selective agonist [125]. LUF5834—2-amino4-(4-hydroxyphenyl)-6-(1H-imidazol-2-ylmethylsulfanyl)pyridine-3,5-dicarbonitrile was described as a partial agonist (EC_50_ of 12 nM for A_2B_ receptor) slightly selective between A_2A_ and A_2B_ (Ki of 28 ± 4 nM and 12 ± 2 nM, respectively) or A_1_ receptor subtypes, but selective over the A_3_ subtype. Its analogue, LUF5835 (2-amino4-(3-hydroxyphenyl)-6-(1H-imidazol-2-ylmethylsulfanyl)pyridine-3,5-dicarbonitrile) is a full A_2B_ agonist with EC_50_ of 10 nM, with a similar selectivity profile. It was later reported that LUF5834 binds to a different receptor site of A_2A_ AR as compared to adenosine-based agonists, suggesting a distinct binding site for this class of agonists on AR receptors [126].

Both of these compounds have yet to be thoroughly characterized in literature (however, they both have been used in cardiac research [127,128]), and have not been proposed as anti-platelet agents.

## 4. Dual Therapy

Anti-platelet therapy is an obvious solution for the treatment and management of arterial thrombosis dependent on blood platelet hyperactivity, often resulting in cardiovascular disease and stroke—the leading causes of morbidity and mortality in developed countries. Several therapeutic options are currently available; however, the problem of efficient and safe therapy remains unsolved, and there is still a demand for novel platelet inhibitors and new therapeutic options.

In clinical practice, efficient anti-platelet treatment is often hindered by reduced sensitivity to many anti-platelet agents. High dosages of anti-platelet drugs, while preventing excessive clotting, frequently also lead to bleeding incidents and moderate to severe side effects. To avoid higher drug doses, combined therapy based on the administration of two or more drugs acting on different platelet activation pathways is often used as an alternative. An example of such an approach currently used in clinical practice is the combined administration of acetylsalicylic acid (an inhibitor of thromboxane A_2_ formation) and clopidogrel (an inhibitor of the P2Y_12_ receptor). The P2Y_12_ receptor is the main therapeutic target in anti-platelet therapy, targeted at the ADP-dependent activation pathway [2]. Its agonization enhances the process of platelet aggregation initiated through the P2Y_1_ receptor. Unfortunately, such treatment is still beset by the problem of resistance, especially among patients with type 2 diabetes, i.e., a group at higher risk of thromboembolic events [129,130,131].

Our research group has recently proposed a novel approach based on the simultaneous application of two anti-platelet agents, a P2Y_12_ antagonist and an AR agonist, which has been found to deepen the action of P2Y_12_ antagonist [110]. Based on this report, we believe that adenosine receptor agonists could significantly enhance the anti-platelet effect of P2Y_12_ antagonists, despite possessing different selectivity profiles and anti-platelet activities. A strategy focused on a purinergic pathway and involving low-dose inhibition of classical (P2Y_12_) purinergic ADP receptors with the simultaneous activation of adenosine receptors may present a novel, promising approach to prevent thrombotic events, and should be further investigated.

## 5. Conclusions

Adenosine receptor agonists have been shown to have anti-platelet effect; however, not all of them are of the same magnitude, with some even presenting no discernible impact on aggregation. It is difficult to unambiguously give a simple answer as to whether this group of compounds stands a fair chance of becoming anti-platelet drugs in the foreseeable future. Too few known AR agonists have been evaluated specifically for modulation of platelet function, and as this topic was studied in the 1980s, some of the data require replication and confirmation using modern methodology. However a few AR agonists, like NECA, HE-NECA, CGS 21680, 2-chloroadenosine, and recently, PSB-15826, were confirmed to have platelet inhibiting properties, and the concept of employing them in preventing thrombus formation is re-emerging. More studies of different AR agonists focused specifically on anti-platelet properties are needed, as predictions based on physicochemical properties prove to be unreliable [68]; however, currently-available data suggests that such attempts should be focused on A_2A_ AR agonists, as agonization of A_2B_ AR has not been reliably reported to impact platelet aggregation or activation.

The use of AR agonists as anti-platelet medication appears feasible following further research focused explicitly on this goal, especially when applied in combination with other anti-platelet agents, to identify therapies demonstrating effective antithrombotic properties without risking severe side effects.

## Figures and Tables

**Table 1 ijms-20-05475-t001:** Adenosine receptors (AR) receptor distribution and expression in different tissue types.

Receptor Subtype	High Expression	Intermediary Expression	Low Expression
A_1_ [19]	brain (cortex, hippocampus, cerebellum); spinal cord; adrenal gland; atria; eyes	brain (excluding cortex, hippocampus, and cerebellum); skeletal muscles; adipose tissue; liver; kidneys	lungs; pancreas
A_2A_ [20]	blood platelets; leukocytes; spleen; thymus	heart; lungs; blood vessels; peripheral nerves	brain
A_2B_ [21,22,23]	cecum; bladder	lungs; blood vessels; mast cells; eyes	brain; adipose tissue; blood platelets; adrenal gland; kidneys
A_3_ [19]	testis; mast cells	brain (hippocampus, cerebellum)	brain (excluding hippocampus and cerebellum); heart; thyroid; adrenal gland; spleen; liver; kidneys

**Table 2 ijms-20-05475-t002:** Nomenclature and chemical structure of AR agonists.

Name	Other Names	IUPAC Name	Structure
2-chloroadenosine	2-Chloro Adenosine, Cl-Ado, 2 ClAdo, 2-CADO	(2R,3R,4S,5R)-2-(6-amino-2-chloropurin-9-yl)-5-(hydroxymethyl)oxolane-3,4-diol	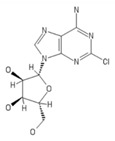
Regadenoson	CVT 3146, CVT-3146, CVT3146, Lexiscan, Rapiscan	1-[6-amino-9-[(2R,3R,4S,5R)-3,4-dihydroxy-5-(hydroxymethyl)oxolan-2-yl]purin-2-yl]-N-methylpyrazole-4-carboxamide	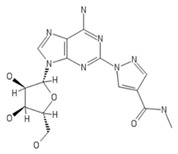
Binodenoson	2-((Cyclohexylmethylene) hydrazino)adenosine	(2R,3R,4S,5R)-2-{6-amino-2-[(E)-2-(cyclohexylmethylidene)hydrazin-1-yl]-9H-purin-9-yl}-5-(hydroxymethyl)oxolane-3,4-diol	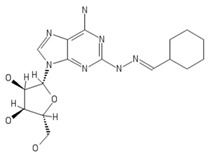
PSB-0777	PSB0777	4-[2-[(6-Amino-9-b-D-ribofuranosyl-9H-purin-2-yl) thio]ethyl]benzenesulfonic acid ammonium salt	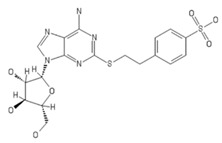
PSB-15826	-	(2S,3S,4R,5R)-5-(6-Amino-2-((2-(4-(4-fluorophenyl)piperazin-1-yl) ethyl)thio)-9H-purin-9-yl)tetrahydrofuran-2,3,4-triol	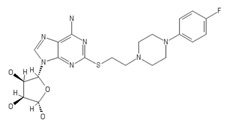
PSB-12404	-	(2R,3R,4S,5R)-2-(6-Amino-2-(2-cyclohexylethylthio)-9Hpurin-9-yl)-5-(hydroxymethyl)tetrahydrofuran-3,4-diol	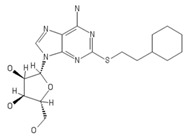
PSB-16301	-	(2S,3S,4R,5R)-5-(6-amino-2-(phenethylthio)-9H-purin-9-yl)tetrahydrofuran-2,3,4-triol	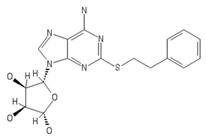
MRE0094	Sonedenoson, 2-[2-(4-Chlorophenyl)ethoxy]adenosine	(2R,3R,4S,5R)-2-[6-amino-2-[2-(4-chlorophenyl)ethoxy]purin-9-yl]-5-(hydroxymethyl)oxolane-3,4-diol	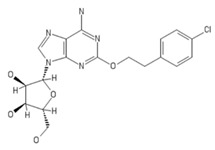
CV1808	2-phenylaminoadenosine, CV-1808	(2R,3R,4S,5R)-2-(6-amino-2-anilinopurin-9-yl)-5-(hydroxymethyl)oxolane-3,4-diol	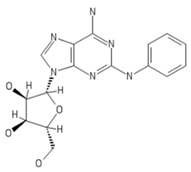
AMP597	RPR 100579	(1S,2R,3S,4R)-4-(4-(((R)-1-(3-chlorothiophen-2-yl)butan-2-yl)amino)-7H-pyrrolo [2,3-d]pyrimidin-7-yl)-N-ethyl-2,3-dihydroxycyclopentane-1-carboxamide	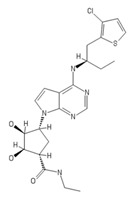
NECA	N-Ethyl-5’-Carboxamido Adenosine, 5’-ethylcarboxamidoadenosine	(2S,3S,4R,5R)-5-(6-aminopurin-9-yl)-N-ethyl-3,4-dihydroxyoxolane-2-carboxamide	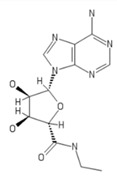
CGS21680	CGS-21680, Cgs 21680, 2-(4-(2-carboxyethyl)phenethylamino)-5’-N-ethylcarboxamidoadenosine	3-[4-[2-[[6-amino-9-[(2R,3R,4S,5S)-5-(ethylcarbamoyl)-3,4-dihydroxyoxolan-2-yl]purin-2-yl]amino]ethyl]phenyl]propanoic acid	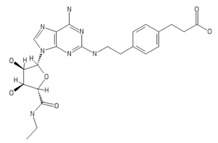
HE-NECA	HENECA, Heneca, 2-hexynyl-NECA, 2-hexynyladenosine-5’-N-ethylcarboxamide	(2S,3S,4R,5R)-5-(6-amino-2-hex-1-ynylpurin-9-yl)-N-ethyl-3,4-dihydroxyoxolane-2-carboxamide	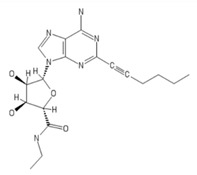
UK-432097	UK-432,097	6-(2,2-diphenylethylamino)-9-[(2R,3R,4S,5S)-5-(ethylcarbamoyl)-3,4-dihydroxyoxolan-2-yl]-N-[2-[(1-pyridin-2-ylpiperidin-4-yl)carbamoylamino]ethyl]purine-2-carboxamide	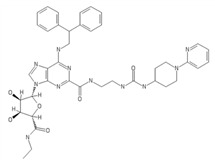
BAY 60-6583	BAY-60-6583, BAY60-6583, 2-((6-amino-3,5-dicyano-4-(4-(cyclopropylmethoxy)phenyl)pyridin-2 yl) sulfanyl)acetamide	2-[6-amino-3,5-dicyano-4-[4-(cyclopropylmethoxy)phenyl]pyridin -2-yl]sulfanylacetamide	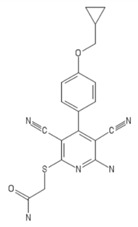
LUF5834	LUF 5834, LUF-5834, 2-Amino-4-(4-hydroxy-phenyl)-6-(1H-imidazol-2-ylmethylsulfanyl)-pyridine-3,5-dicarbonitrile	2-amino-6-(1H-imidazol-2-ylmethylsulfanyl)-4-(4-oxocyclohexa-2,5-dien-1-ylidene)-1H-pyridine-3,5-dicarbonitrile	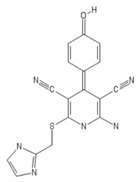
LUF5835	LUF 5835, LUF-5835	2-amino-6-(1H-imidazol-2-ylmethylsulfanyl)--4-(3-hydroxy-phenyl) pyridine-3,5dicarbonitrile	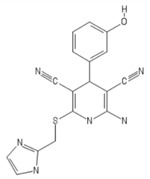

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
