# Peer review of "Blood Platelet Adenosine Receptors as Potential Targets for Anti-Platelet Therapy"

_ijms, 2019, doi:10.3390/ijms20215475_

Round 1

Reviewer 1 Report

Wolska and Rozalski have submitted a useful review on the current status in the field of platelet adenosine receptors (ARs), current analogues for ARs that are being studied and if any of these have made it to clinical trials.

This reviewer likes the showing to the AR analogue structures and details on binding affinities, but would like to see a figure/model of platelet adenosine receptor signalling and roles in platelet aggregation to help in the understanding of the AR receptors for the non-platelet/ non-adenosine receptor readership, in what otherwise is a fairly comprehensive review.

It would also be great if there was some more detail on the current bleeding risks of the P2Y12 antagonists used currently, to help further support the need for research in AR antagonism.

Although there is some mention of the expression levels of the AR receptors p2 line 60+, it would be good to see how/mention how these levels of expression related to P2Y12 receptors and other major platelet receptors that are used as current therapeutic targets.

The sections on the derivatives are fairly detailed, but would be good to have a summary table of derivate’ binding affinities, IC50s/EC50s and Ki.

Minor points:

P2 line 48 – “superior clinical benefits” of use of prasugrel and ticagrelor, would be good to mention some of these not just reference a paper.

P3 Line 95 – Could do with a reference. It does come further down but would be good to reference when first mentioning the study and main finding. Likewise in Line 103.

P7 Line 143 – define PRP for non-platelet readership

P7 – When mentioning the binding affinities and Ki values for the derivatives, would be good to have both in all sections. Some derivatives have mentions of both, others just one or the other. Perhaps a table of binding affinities and Kis would help the reader.

P12 – References

Some references are not consistent. i.e. using short titles for journals and formatting of page numbers. Sometimes pp., sometimes just numbers. Some references also contain full author list, whilst some have three and et al.

Author Response

Answers to the comments and points made by the Reviewers

Reviewer 1

This reviewer likes the showing to the AR analogue structures and details on binding affinities, but would like to see a figure/model of platelet adenosine receptor signalling and roles in platelet aggregation to help in the understanding of the AR receptors for the non-platelet/ non-adenosine receptor readership, in what otherwise is a fairly comprehensive review.

A model of platelet adenosine receptor signalling and its role in the platelet activation/aggregation is presented in the form of Graphical Abstract. We believe that presenting this model once again as the figure in the text would be a redundancy. We can however transform the graphical abstract into the figure in the manuscript if it is recommended by the Editor.

It would also be great if there was some more detail on the current bleeding risks of the P2Y12 antagonists used currently, to help further support the need for research in AR antagonism.

As suggested, the additional information has been added in the Introduction in the corrected version of the manuscript:

“For example, gastrointestinal bleeding is a common adverse event observed in 5 to over 10% of patients treated with oral antiplatelet drugs. Many of the patients with this complication require the recurrent hospitalization [8]. Another severe and relatively common side effect of antiplatelet therapy is a higher risk of intracranial and intracerebral haemorrhage [9]. On the other hand, for some patients, the applied antiplatelet therapy appears insufficient and does not prevent excessive clotting.”

Although there is some mention of the expression levels of the AR receptors p2 line 60+, it would be good to see how/mention how these levels of expression related to P2Y12 receptors and other major platelet receptors that are used as current therapeutic targets.

As it was suggested by the Reviewer 1, additional information has been added in “The classification, distribution and signalling of adenosine receptors” section in the manuscript.

The paragraph now is as follows (changes have been highlighted):

“As it has been already mentioned, blood platelets express two subtypes of AR receptors (A2A and A2B) however the expression (the number of receptor copies in the plasma membrane) of the two receptor types has not been established. A2A receptor is believed to be expressed on platelets in higher density as compared to A2A [18]. Only one study has estimated the gene expression profile for A2A and A2B in human platelets. This report demonstrated comparable mRNA expression levels for A2A and A2B AR [26], but no further evidence exists regarding protein levels present on the platelet surface. Moreover, two studies [27,28] have not been able to quantify A2A AR in platelet proteome, while easily identified for example P2Y12 receptor, expressed on healthy platelets in around 450–1,000 copies [29].”

The sections on the derivatives are fairly detailed, but would be good to have a summary table of derivate’ binding affinities, IC50s/EC50s and Ki.

IC50s/EC50s measured for AR agonists are highly dependent on conditions used to establish those measures, therefore it is not possible to compare them directly. We have added the following information in the corrected version of the manuscript:

It is however important to point out that citied study was performed using various protocols and utilising nonhomogeneous materials, hence is not possible to make a direct comparison between available data, for example for IC50 or EC50 measured for various AR agonists.

P2 line 48 – “superior clinical benefits” of use of prasugrel and ticagrelor, would be good to mention some of these not just reference a paper.

According to the suggestion of the Reviewer 1, we have added some information in the corrected version of the manuscript. The sentence now is as follows:

Although all three agents have an indication for use in acute coronary syndromes, current guidelines suggest the preferential use of prasugrel and ticagrelor over clopidogrel because of their superior clinical benefits, i.e. the improved efficacy, lowered individual variation in response, less frequent and severe side effects [6].

P3 Line 95 – Could do with a reference. It does come further down but would be good to reference when first mentioning the study and main finding. Likewise in Line 103.

As suggested, we have inserted the references “earlier” in the both paragraphs and also we have added more references there.

P7 Line 143 – define PRP for non-platelet readership

Definition (PRP – platelet rich plasma) has been included in the corrected version of the manuscript

P7 – When mentioning the binding affinities and Ki values for the derivatives, would be good to have both in all sections. Some derivatives have mentions of both, others just one or the other. Perhaps a table of binding affinities and Kis would help the reader.

For each AR agonist we tried to include all available data, however some data (binding affinities or Ki values) regarding A2A and A2B AR receptors have not been established and reported in literature for certain agonists.

P12 – References

Some references are not consistent. i.e. using short titles for journals and formatting of page numbers. Sometimes pp., sometimes just numbers. Some references also contain full author list, whilst some have three and et al.

We have corrected and unified the References and used short titles for journal names. As regards the author list, we have followed a rule from the Instruction to Authors of IJMS: “For documents co-authored by a large number of persons (more than 10 authors), you can either cite all authors, or cite the first ten authors, then add a semicolon and add ‘et al.’ at the end: Author 1; Author 2; Author 3; Author 4; Author 5; Author 6; Author 7; Author 8; Author 9; Author 10; et al.”  

Reviewer 2 Report

In this well written manuscripts the authors describe their hypothesis that the use of adenosine receptor agonists might be beneficial additive treatment for platelet aggregation inhibition. The authors review some of the current adenosine derivatives and their effects on platelets.

Major concerns

The authors should discuss the potential adverse effects of increased adenosine receptor stimulation, as well as increased extra- and intra- cellular concentrations of adenosine analogues. This discussion should also related to the current “experiment of nature” of patients with excess adenosine, i.e those suffering from adenosine deaminase deficiency. Moreover, the authors should comment on the effects of this deficiency on platelet function.

Minor concerns

Line 123: change “ether” to “either”

Table 2: The authors should clarify the need for table 2. Alternatively, it could be moved to a supplemental file.

Author Response

Answers to the comments and points made by the Reviewers

Reviewer 2

The authors should discuss the potential adverse effects of increased adenosine receptor stimulation, as well as increased extra- and intra- cellular concentrations of adenosine analogues. This discussion should also related to the current “experiment of nature” of patients with excess adenosine, i.e those suffering from adenosine deaminase deficiency. Moreover, the authors should comment on the effects of this deficiency on platelet function.

As it was suggested by the Reviewer 2, we have added this information in the corrected version of the manuscript in the section: “2. The classification, distribution and signalling of adenosine receptors”:

“Adenosine receptors play multiple functions. A2A receptor agonization is known to cause coronary artery vasodilatation, decreased dopaminergic activity in central nervous system, inhibition of central neuron excitation, whereas A2B receptor activation may cause bronchospasm [12,19-21]. Therefore, all adenosine receptor ligands should be used only with outmost caution [22], despite some have already been approved for human use (one of them – regadenoson is discussed further in this article). An interesting insight into the adenosine receptor overstimulation may be gained from the study of adenosine deaminase deficiency – a rare, autosomal metabolic disorder that causes severe combined immunodeficiency [23]. In this syndrome platelet dysfunction has been described, as well as severe thrombocytopenia [24,25]. It is however important to remember that there is no exact parallel between patients with this syndrome and an antiplatelet therapy with the use of AR agonists. The dose of synthetic adenosine agonist equivalent to adenosine would be much lower, and most importantly it would be applied in adults..”

Line 123: change “ether” to “either”

It has been corrected in the newer version of the manuscript.

Table 2: The authors should clarify the need for table 2. Alternatively, it could be moved to a supplemental file.

We believe that Table 1 that presents full names, alternative names used in literature and chemical structures of AR agonists can be helpful for the reader. However, we could move the Table to supplementary material if it is suggested by the Editor.